# Risk of New Primary Cancer in Patients with Posterior Uveal Melanoma: A National Cohort Study

**DOI:** 10.3390/cancers14020284

**Published:** 2022-01-07

**Authors:** Mette Bagger, Vanna Albieri, Tine Gadegaard Hindso, Karin Wadt, Steffen Heegaard, Klaus Kaae Andersen, Jens Folke Kiilgaard

**Affiliations:** 1Department of Ophthalmology, University Hospital of Copenhagen, 2100 Copenhagen, Denmark; tine.gadegaard.hindsoe@regionh.dk (T.G.H.); steffen.heegaard@regionh.dk (S.H.); jens.folke.kiilgaard@regionh.dk (J.F.K.); 2Statistics and Data Analysis, Danish Cancer Society Research Center, 2100 Copenhagen, Denmark; albieri.vanna@gmail.com (V.A.); klaus.kaae.70@gmail.com (K.K.A.); 3Department of Clinical Genetics, University Hospital of Copenhagen, 2100 Copenhagen, Denmark; karin.wadt@regionh.dk; 4Department of Pathology, University Hospital of Copenhagen, 2100 Copenhagen, Denmark

**Keywords:** new primary cancer, uveal melanoma, cancer incidence, epidemiology

## Abstract

**Simple Summary:**

Prior studies on the risk of new primary cancer among patients with posterior uveal melanoma have produced conflicting results, and the role of other risk factors relevant to cancer formation, such as socioeconomic status, has not been investigated. The focus on the genetic susceptibility of cancer among patients with uveal melanoma has increased with the recognition of BRCA1-Associated Protein 1 (BAP1) tumor predisposition syndrome presenting with an increased incidence of uveal melanoma, renal cell carcinoma, mesothelioma, and cutaneous melanoma in the affected family members. Our study evaluates the risk of new primary cancer in a validated almost complete national cohort of clinically and histopathologically well described posterior uveal melanomas from 1968 through 2016. Our study showed a 21% increased incidence of new primary cancer following the diagnosis of posterior uveal melanoma. The risk was independent of socioeconomic factors and was not restricted to specific cancer types.

**Abstract:**

Background: Studies on the risk of new primary cancer in patients with posterior uveal melanoma (UM) have produced conflicting results, and the role of socioeconomic status (SES) is unknown. The purpose of this population-based matched cohort study was to determine the risk of new primary cancer following the diagnosis of posterior UM. Methods: 2179 patients with posterior UM 1968–2016 and 22,717 matched controls without cancer were included. Incidence and time-dependent hazard ratio (HR) of new primary cancer were described, and the effect of SES was emphasized in a sub-cohort. Results: The incidence of new primary cancer was increased in patients with posterior UM, rate ratio (RR) 1.21 (95% CI: 1.08; 1.35), but the specific cancer types did not differ compared to the controls. The rate of new primary cancer following the diagnosis of posterior UM was significantly increased 2–5 years (HR 1.49 (95% CI: 1.23; 1.80)) and 11–15 years (HR: 1.49 (95% CI: 1.12; 1.99)), and adjusting for SES did not change the rate (HR 1.35 (95% CI:1.20; 1.55)). Conclusions: Patients with posterior UM have an increased risk of new primary cancer independent of SES. No difference in incidence of specific cancer type was observed compared to the control group.

## 1. Introduction

Uveal melanoma (UM) comprises approximately 5% of all melanomas and arises from melanocytic cells in the uveal tract [1]. The UM cells can either be classified as spindle, epithelioid or mixed [2]. UM can be located anteriorly in the iris or posteriorly in the ciliary body and the choroid [2]. Iris melanomas are a less common and distinct subtype of UM which rarely metastasize, while posterior UM have a 45% disease-specific mortality by 15 years [3]. Dissemination from posterior UM can occur several decades after treatment of the primary tumor [3]. The prognosis of posterior UM can be estimated with the American Joint Committee on Cancer (AJCC) staging system [4], which includes the largest basal diameter, tumor height, involvement of ciliary body and the presence of extra scleral extension [5]. Acquired genetic alterations, especially loss of chromosome 3 and gain of chromosome 8q, are also important factors for the risk of metastatic disease [2]. The suspicion of shared genetic susceptibility to the development of second primary cancer among patients with UM has been emphasized by the recent identification of the BRCA1-Associated Protein 1 (*BAP1*) tumor predisposition syndrome presenting with a pathogenic germline variation in the *BAP1* tumor suppressor gene, which is located on chromosome 3, and an increased incidence of uveal melanoma, renal cell carcinoma, mesothelioma, and cutaneous melanoma in the affected family members [6,7]. Currently, germline *BAP1* variations have only been identified in a few Danish families and 181 families worldwide [8,9]. An association between hereditary predisposition to uveal melanoma and germline pathogenic variants in other known tumor genes such as *MBD4*, *PALB2,* SMARCE1 and *MLH1* has also been suggested [10].

Most follow-up programs for patients with posterior uveal melanoma focuses mainly on the liver with biannual ultrasonography, as the liver is the first site of distant metastases in 93% of cases [11]. There are to date no evidence-based guidelines in regard to the surveillance of new primary cancer during the follow-up of patients with posterior UM, and previous studies evaluating the incidence of new primary cancer have so far produced conflicting results [12,13,14]. We therefore conducted a Danish population-based cohort study to examine the risk of new primary cancer among patients with posterior UM compared to matched individuals with no prior history of cancer. We also assessed if socioeconomic status had a significant influence on the risk of new primary cancer among the patients with posterior UM and if the frequency of specific cancer types differed from the matched comparisons.

## 2. Materials and Methods

### 2.1. Study Population

The study included Danish patients diagnosed with posterior UM from 1968 to 2016. Several resources were used to identify and validate cases, including review of all eye-pathology reports since 1968, clinical charts, the Danish Cancer Registry (DCR), and the Danish Ocular Oncology Group (DOOG) database. The identification and validation procedure of patients with posterior UM have previously been described in detail [15]. Every Danish citizen has had a unique 10-digit personal identification (CPR) number since 1968, which allows unambiguous individual-level data cross-linkage [16]. Date of birth, date of death and migration were available from the Danish Civil Registration System through linkage to the CPR number [16]. Vital status was extracted by the end of the data collection period on 31st December 2018 [16]. The CPR number was used to link data on an individual level from the Danish Cancer Registry (DCR). All patients with posterior UM with a previous cancer diagnosis were excluded from the study, to remove the potential effect of treatment interference and diagnostic bias [17]. Ten individuals matched on sex and date of birth were randomly sampled from the Danish Civil Registration System for every study case. The unexposed comparisons were alive, living in Denmark and not diagnosed with posterior UM or any other cancer at the date of diagnosis of the appertaining patient with posterior UM. In cases of posterior UM among the unexposed comparisons, the follow-up was terminated, and the individual was placed in the cohort of study cases. Causes of death were obtained through the Registry of Causes of Death [18].

The study was conducted in accordance with the tenets of the Declaration of Helsinki. The Regional Research Ethical Committee in Copenhagen waived the need for approval of this retrospective study (protocol no: H-4-2014-FSP). Collection and linkage of data was approved by the Danish Data Protection Agency (protocol no: 2016-41-4897) and the Danish Health Authority (protocol no: 3-3013-727). The manuscript was prepared in accordance with the STROBE statements for cohort studies.

### 2.2. Study Outcomes 

Posterior UM patients and unexposed comparisons were followed from their diagnosis/index date and the diagnosis of a subsequent cancer. Incidence of new primary cancer except non-melanoma skin cancer was extracted from the DCR. The validity and a high degree of completeness of the DCR are secured through manual quality control routines, application of automatic cancer logic and use of information from multiple different data sources [17]. However, metastasis from posterior UM has frequently been coded as a new diagnosis of metastatic cutaneous melanoma (with no prior history of primary cutaneous melanoma). Thus, to mitigate this systematic coding error, we excluded all reported cases of cutaneous melanoma with distant metastases at the time of diagnosis. Follow-up continued until the date of new primary cancer, death, emigration, or end of study date, whichever came first. 

### 2.3. Statistical Methods 

Raw incidence rates (IRs), incidence risk ratios (RRs), and absolute excess rate (AER) with 95% confidence intervals were estimated to evaluate crude measures for new primary cancer in the posterior UM patients with respect to comparisons.

Cumulative incidence function was employed to describe the incidence of subsequent cancer to account for the effect of competing events of death (all-cause) due to the increased mortality among patients with posterior UM, with time since diagnosis/index date used as an underlined timescale. Cumulative incidence curves were constructed for patients with posterior UM and unexposed comparisons. The study subjects were additionally stratified for AJCC stage (levels: stage I, stage II, stage III–IV, comparisons) [4]. Gray’s test and the Fine–Gray regression model were used to evaluate the effect of patients with posterior UM compared to unexposed comparisons on incidence of new primary cancer with all-cause death as a competing event. 

Cox proportional hazard models were applied to estimate hazard ratios (HRs), with 95% confidence intervals (CI), for subsequent cancer among posterior UM patients with respect to comparisons, adjusting for age at cancer/index date (levels: 0–19, 20–39, 40–49, 50–59, 60–69, 70–79, 80+), calendar year at cancer/index date (levels: 1968–1979, 1980–1989, 1990–1999, 2000–2009, 2010–2016) and gender. Information on socioeconomic position is available on Danish residents starting 1st January 1980. Thus, in a sub-cohort study with posterior UM patients and comparisons diagnosed/indexed 1980–2016, we further adjusted for civil status (levels: single, in relationship), education (levels: education levels: long, intermediate, short and unknown) and income (adjusted quintiles by calendar year, age and gender on the entire Danish population). Proportional hazard assumption was evaluated by testing the correlation coefficient for transformed time and scaled Schoenfeld residuals and associated plots [10]. All analysis was conducted using the statistical software R (version 3.5.1, R Core Team, Vienna, Austria, and packages ‘survival’, ‘etm’, ‘cmprsk’, and ‘riskRegression’) [11].

## 3. Results

A total of 2179 patients with posterior UM, diagnosed from 1968 through to 2016 and with no prior history of cancer, were included in the study and followed for the incidence of new primary cancer. Baseline characteristics among patients and matched comparisons are listed in Table 1. In a sub-cohort of patients diagnosed with posterior UM from 1980 to 2016, we also included socioeconomic factors. Among 1520 (69.8%) deaths in patients with posterior UM, 598 (27.4 %) deaths were related to cancer. Comparisons experienced a total of 12,529 (55.2) deaths, of which 1875 (8.3%) were cancer related. Patients with posterior UM had a follow-up of 19,585 person-years, with a median of 5.9 person-years (interquartile range (IQR): 9.7), and comparisons had a follow-up of 317,321 person-years, with a median of 11.6 person-years (IQR: 14.0) (Table 2). The registry of causes of death did now allow for a valid distinction between death from posterior UM and death from other primary cancers. Only 262 (1.2%) in the comparison group and 15 (0.7%) in the posterior UM patient group were lost to follow-up due to migration. At the end of the study, 9926 (43.7%) and 644 (29.6%) were alive and censored.

### 3.1. Incidence of New Primary Cancer

For new primary cancer outcome, patients with posterior UM had a total follow-up of 18,697 person-years, with a median of 5.6 person-years (IQR: 9.3), and comparisons had a total follow-up of 303,620 person-years, with a median of 11.0 person-years (IQR: 13.7). The crude incidence rates of new primary cancer among patients with posterior UM and comparisons are listed in Table 2. The results show a 21% (RR 1.21 (95% CI 1.08–1.35) increased incidence of new primary cancer among patients with uveal melanoma compared to the matched comparisons (Table 2). The excess risk was 2.99 cases of new primary cancer per 1000 person-years of follow-up among the patients with posterior UM compared to the comparisons (absolute excess rate (AER) 2.99 (95% CI 1.06–4.93)) (Table 2).

The cumulative incidence curves of new primary cancer, with death from all causes as a competing event, showed a tendency for an increased incidence during the first decade, and thereafter a decreased incidence for patients with posterior UM related to the controls (Figure 1A). The Grey’s test confirmed the significant difference between the two groups on the cumulative incidence for new primary cancer (*p*-value < 0.001). As expected, the all-cause mortality was significantly higher among patients with posterior UM (Figure 1B). We stratified the cumulative incidence curves for patients with posterior UM according to AJCC stage. The incidence of new primary cancer was significantly elevated among patients with AJCC stage I tumors (Figure 1C). Surprisingly, the cumulative all-cause mortality among patients with a stage I tumor was comparable to the mortality of the controls. The tendency of an increased risk of new primary cancer was also present for patients with AJCC stage III–IV tumors but the analysis did not detect an increased incidence of new primary cancer in this group due to the high mortality (Figure 1C). 

### 3.2. Incidence of Specific Cancer Types

After the exclusion of all cases of cutaneous melanoma with a simultaneous code for metastatic disease, there were only 154 and 7 cases of malignant cutaneous melanoma among comparisons and posterior UM patients, respectively. This corresponded to an incidence rate of 0.05 (95% CI: 0.04; 0.06) and 0.4 (95% CI: 0.02; 0.08) per 100 person-years of follow-up for comparisons and patients with posterior UM, respectively, and a rate ratio of 0.74 (95% CI: 0.35; 1.57). Thus, we found no increased incidence of cutaneous melanoma. There were less than five cases of mesothelioma and renal cell carcinoma. Thus, we did not detect an increased risk of the cancers which are associated to the BAP1 tumor predisposition syndrome [9]. The frequency of cancer type is shown in Table 3 and we found no difference in the occurrence of specific cancers among comparisons and posterior UM patients (Table 3). 

### 3.3. Hazard Ratio of New Primary Cancer

The hazard ratio for new primary cancer varied over time with a peak of increased rate at 2–5 years after diagnosis (Figure 2). We therefore presented the hazard ratio for new primary cancer in patients with posterior UM with respect to unexposed comparisons as a time-dependent variable even though the proportionality assumption was not violated. The estimates showed a trend towards an increased rate throughout the entire follow-up period (Table 4). The Fine–Gray model for the new primary cancer, considering death (all-cause) as a competing event, showed for the first 10 years after diagnosis/index date an increased incidence for patients with posterior UM with respect to the unexposed comparisons, but then the incidence decreased (results not shown). In a sub-cohort of patients with posterior UM and comparisons diagnosed/matched from 1980 until 2016, the adjustment for education, income, and civil status did not affect the estimated hazard ratio of new primary cancer among patients with posterior UM with respect to the unexposed comparisons (Appendix A), HR 1.37 (95% CI: 1.20; 1.55).

## 4. Discussion

Overall, our population-based matched cohort study found an increased risk of new primary cancer, independent of socioeconomic status, among patients with posterior UM compared to a matched cohort of individuals with no prior history of cancer. The unique registries in Denmark allowed for a direct comparison of patients diagnosed with posterior UM and individuals from the background population, which were matched 1:10 on gender and date of birth and entered the study at time of diagnosis of the corresponding patient with posterior UM. We found no difference in the pattern of specific cancer sites between posterior UM patients and comparisons.

Previous studies evaluating new primary cancers in uveal melanoma have used standardized incidence rates, which also provides a representable sample of the background population regarding gender, age and calendar period, but our matched design had the additional advantage that the two cohorts did not overlap [13]. Our study was the first to include the role of socioeconomic status, which has been shown to influence the risk of several cancers, including lung cancer, female breast cancer and prostate cancer [19]. Interestingly, the socioeconomic status had no effect on the rate of a new primary cancer among patients with posterior UM. This could indicate that the cancer development is primarily driven by a genetic vulnerability in this patient group. At present, we only know the *BAP1* tumor predisposition syndrome, and this syndrome comprises only an insignificant fraction of the patients with accumulated cancers in our series, in accordance with a 3% occurrence of BAP1 variants in patients with posterior UM [9]. Whole-exome sequencing of 27 patients with familial UM identified other potential pathogenic variants in PALB2 (one case), SMARCE1 (one case) and MLH1 (one case) [10]. However, the role of these established cancer genes in uveal melanoma patients needs to be further investigated.

Our findings emphasize increased awareness of malignant comorbidity in these patients, as early diagnosis and treatment of new primary cancers can be crucial for improved survival. This also advocates for a histopathological diagnosis of metastases to ensure the diagnosis of metastatic uveal melanoma, as treatment protocols depend critically on tumor type. A meta-analysis study highlighted that cancer survivors generally receive more frequent cancer-screening compared to matched controls, but the survival benefit of this increased screening tendency has not yet been demonstrated [20]. The use of PET-CT for follow-up of patients with uveal melanoma has been shown to provide detection of new undiagnosed primary cancers in 4.3% of cases [21]. The study suggested a potential benefit of PET-CT for early detection of new neoplasms, but the evidence was not sufficient to justify intensified surveillance for all patients with posterior UM [21]. Characterization of the genetic profile in patients with new primary cancer might aid the selection of future patients who could benefit from intensified follow-up with PET-CT; however, this was beyond the scope of this study. 

Our findings of an increased risk were consistent with a Swedish population study, based on the Swedish Cancer Registry, and two cohort studies, based on data from the Surveillance Epidemiology and End Results (SEER) cancer registry [12,13,22]. A previous Danish cohort study from 1995 identified an increased rate of new cancer among males only, but the study relied on registry data only and included all patients with an ocular cancer rather than only patients diagnosed with posterior UM [23]. Our study was based on a validated cohort of clinical and histopathological well described patients with clinically and histopathologically well described posterior uveal melanomas; thus, we avoided the risk of including other ocular tumors, which was miscoded as posterior UM. Other studies including the prospective cohort study performed by the Collaborative Ocular Oncology group also included well described study subjects but did not report increased rates of new primary cancer among uveal melanoma patients. This could be due to the study designs which did not allow for a representable unexposed comparison group. Instead, the COMS study used the expected incidence of cancer in the general population as a reference [24,25].

Initial PET-CT scans on consecutive patients diagnosed with posterior UM have identified new synchronous cancers in 3.3% of cases [26]. To subtract the detection of other synchronous cancers during the initial work-up of the patients diagnosed with posterior UM, we analyzed the risk during the first year separately. Interestingly, the risk of new primary cancer was elevated in the posterior UM group only after 2–5 years, indicating that the increased risk in posterior UM patient was not due to detection bias. After five years, we observed a decrease in the risk among the posterior UM group, and we speculate that the decrease was caused by excessive mortality among posterior UM patients. This might suggest that the patients with the worst prognosis also had the highest risk of new primary cancer and emphasize a shared genetic susceptibility for the development of new primary cancer and posterior UM with adverse genetic aberrations causing a poor prognosis.

The validity and completeness of our primary cohort of patients with posterior UM was ensured by review of all pathology reports since 1968, available patient charts and cross-linking to the Danish Cancer Registry as previously described [15]. It also allowed us to evaluate only patients with posterior UM and exclude patients with iris melanoma which represent a different sub-class of melanoma with distinct clinical and epidemiological characteristics [27,28,29]. We observed a systematic error in the Danish Cancer Registry, where metastatic uveal melanoma was misclassified as primary cutaneous melanoma with distant metastases. This issue has been previously described by other registry-based studies [13,22]. Thus, in order to avoid coding bias, we excluded all cutaneous melanoma cases which presented with a combined coding of new cutaneous melanoma and metastatic disease. We acknowledge that we could potentially miss some cases of stage IV cutaneous melanoma. Once the assumed misclassified cases were removed from the analyses, we no longer detected an increased incidence of cutaneous melanoma among the patients with posterior UM. 

As expected, there was an excess mortality among the patients with uveal melanoma compared to the unexposed comparisons. Thus, the person-time at risk was limited for patients with posterior UM compared to the matched comparisons. This could explain why the increased rate of new primary cancer diminished beyond 10 years of follow-up in the cumulative incidence function (Figure 1A). As opposed to previous studies, our cohort of clinically well described posterior UM allowed for the stratification of patients according to AJCC stage [13,14,24]. This provided a unique opportunity to evaluate the incidence of a new primary cancer in a subgroup of patients with stage I tumors, where the cumulative incidence of death was similar to the comparisons (Figure 1D). Interestingly, the incidence of new primary cancer among patients with the same mortality as the comparisons remained significantly increased throughout the study period (Figure 1C), while increased incidence of new primary cancer could not be demonstrated in stage II and stage III–IV tumors, possibly due to high mortality and, consequently, short follow-up. This issue was investigated by Cronin Fenton et al. [30], who showed that patients with prostate cancer had a much lower accumulated person time compared to their matched control group. This caused a progressively higher age distribution among the matched control group, causing an underestimation of the cancer incidence among the prostate cancer patients [30].

## 5. Conclusions

Patients with posterior UM have an increased risk of new primary malignancies independent of socioeconomic status. We did not identify the increased risk to depend on specific cancer types. Our results underline the importance of an increased awareness for early detection of other primary malignancies during the follow-up of patients with posterior UM. In the event of suspected metastatic lesions, histopathological confirmation should be considered in order to not miss the diagnosis of a new primary cancer and the possibility of potential beneficial treatment options.

## Figures and Tables

**Figure 1 cancers-14-00284-f001:**
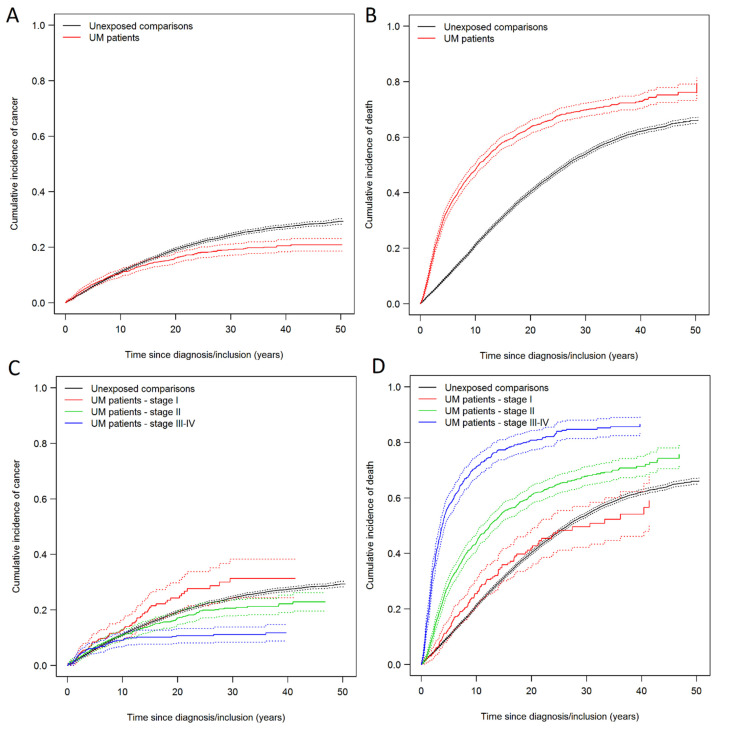
(**A**) Solid lines show the cumulative incidence of new primary cancer (all cancer types except posterior uveal melanoma and non-melanoma skin cancer (NMSC)) with death as a competing event. (**B**) Solid lines show cumulative incidence of death (all-cause). (**C**) Solid lines show cumulative incidence of new primary cancer (all cancer types except posterior UM and NMSC) with death (all-cause) as a competing event. Uveal melanoma patients stratified according to American Joint Committee on Cancer (AJCC) stage. (**D**) Solid lines show cumulative incidence of death (all-cause) stratified according to AJCC stage. Dotted lines shows the 95% confidence intervals.

**Figure 2 cancers-14-00284-f002:**
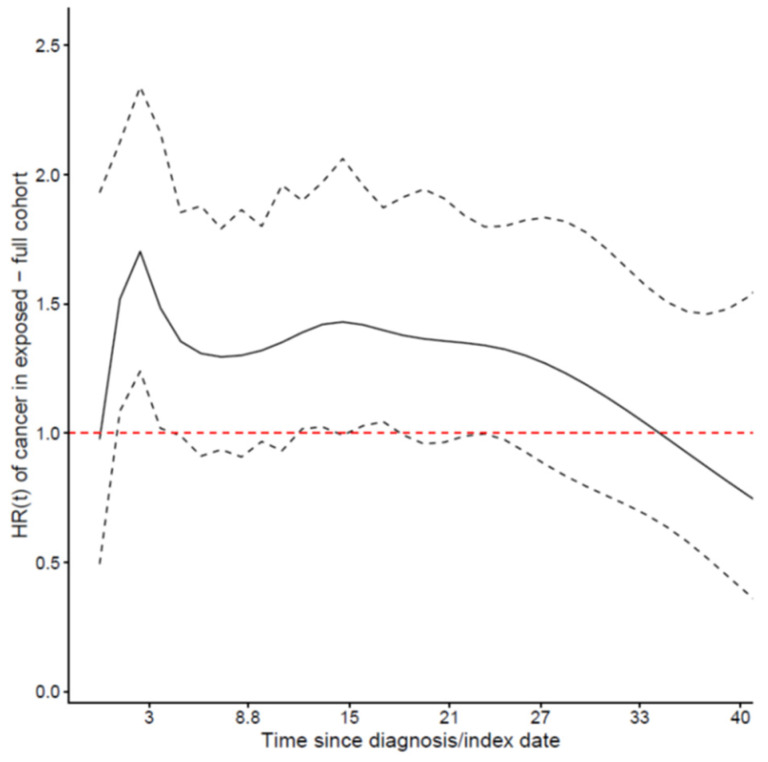
Solid line shows the time-dependent hazard ratio HR(t) of new primary cancer among posterior uveal melanoma patients with respect to the unexposed comparisons in relation to time since diagnosis/index date (years) (adjusted analyses, full cohort). The dotted line shows the 95% confidence intervals.

**Table 1 cancers-14-00284-t001:** Descriptive statistics of full cohort (1968–2016) and sub-cohort (1980–2016).

Levels	Full Cohort		Sub-Cohort (1980–2017)	
	Unexposed ^3^	UM ^2^ Patients	Unexposed ^3^	UM ^2^ Patients
**Total number**	22,717	2179	17,908	1738
Sex (%)				
Male	11,695 (51.5)	1145 (52.5)	9150 (51.1)	908 (52.2)
Female	11,022 (48.5)	1034 (47.5)	8758 (48.9)	830 (47.8)
**Age index (%)**				
<20	90 (0.4)	9 (0.4)	59 (0.3)	6 (0.3)
20–39	1507 (6.6)	164 (7.5)	1089 (6.1)	120 (6.9)
40–49	2418 (10.6)	249 (11.4)	1867 (10.4)	200 (11.5)
50–59	4555 (20.1)	445 (20.4)	3618 (20.2)	359 (20.7)
60–69	6626 (29.2)	624 (28.6)	5235 (29.2)	496 (28.5)
70–79	5421 (23.9)	499 (22.9)	4317 (24.1)	401 (23.1)
≥80	2100 (9.2)	189 (8.7)	1723 (9.6)	156 (9.0)
**Year index (%)**				
1968–1979	4809 (21.2)	441 (20.2)	-	-
1980–1989	4311 (19.0)	395 (18.1)	4311 (24.1)	395 (22.7)
1990–1999	4650 (20.5)	454 (20.8)	4650 (26.0)	454 (26.1)
2000–2009	5005 (22.0)	484 (22.2)	5005 (27.9)	484 (27.8)
2010–2016	3942 (17.4)	405 (18.6)	3942 (22.0)	405 (23.3)
**AJCC ^1^ stage**				
I	-	365 (16.8)	-	309 (18.3)
II	-	1136 (52.1)	-	905 (53.6)
III–IV	-	591 (27.1)	-	473 (28.0)
NA		87 (4.0)		51
**Disposable Income (%)**				
1	-	-	3280 (18.3)	305 (17.6)
2	-	-	3604 (20.1)	355 (20.4)
3	-	-	3782 (21.1)	374 (21.5)
4	-	-	3550 (19.8)	358 (20.6)
5	-	-	3606 (20.1)	345 (19.9)
NA	-	-	86 (0.5)	0 (0.0)
**Civil status (%)**				
Single	-	-	6539 (36.5)	608 (35.2)
Relationship	-	-	11,231 (62.7)	1121 (64.8)
NA	-	-	138 (0.8%)	
**Education (%)**				
Higher	-	-	2776 (15.6)	288 (16.7)
Medium	-	-	6369 (35.8)	665 (38.5)
Short	-	-	4732 (26.6)	424 (24.5)
Unknown	-	-	3921 (22.0)	352 (20.4)

^1^ American Joint Committee on Cancer. ^2^ Posterior uveal melanoma. ^3^ Unexposed comparisons.

**Table 2 cancers-14-00284-t002:** Crude rates of new primary cancer (all cancer types except posterior uveal melanoma and non-melanoma skin cancer).

Subsequent Cancer	Events	PY ^1^	IR ^2^	(95% CI) ^3^	RR ^4^	95% CI ^3^	AER ^5^	(95% CI)
Posterior UM ^6^: New primary cancer	324	18,696.60	1.73	(1.55; 1.93)	1.21	(1.08; 1.35)	2.99	(1.06; 4.93)
Unexp ^7^: New primary cancer	4353	303,620.38	1.43	(1.39; 1.48)				
Death								
Posterior UM: All-cause	1520	19,585.13	7.76	(7.38; 8.16)	1.97	(1.86; 2.07)	38.13	(34.16; 42.09)
Unexp: All-cause	12,529	317,321.09	3.95	(3.88; 4.02)				
Posterior UM: All cancer-related death	598	19,585.13	3.05	(2.81; 3.31)	5.17	(4.71; 5.67)	24.62	(22.16; 27.09)
Unexp: All cancer-related death	1875	317,321.09	0.59	(0.56; 0.62)				

^1^ Person-years, ^2^ Incidence rates, ^3^ 95% confidence interval, ^4^ Rate ratios, ^5^ Absolute excess rates, ^6^ posterior uveal melanoma, ^7^ Unexposed comparisons. IRs and AERs and corresponding 95% CIs are reported per 100 and 1000 person-years, respectively.

**Table 3 cancers-14-00284-t003:** Events of new primary cancer specified on cancer type.

New Primary Cancer Site	Unexposed	Posterior UM Patients
Bone and joints	6 (0.1)	0
Breast cancer	480 (11.0)	34 (10.5)
Oral cavity and pharynx	113 (0.6)	<10 ^1^ (NA)
Digestive organs	1153 (26.5)	83 (25.6)
Endocrine glands	19 (0.4)	0
Eye and CNS	58 (1.3)	<10 ^1^ (NA)
Female genital organs	215 (4.9)	19 (5.9)
Lymphatic/hematologic tissue	287 (6.6)	20 (6.2)
Male genital organs	633 (14.5)	50 (15.4)
Mesothelioma and connective tissue	22 (0.5)	<10 ^1^ (NA)
Respiratory system and intrathoracic organs	709 (16.3)	41 (12.7)
Skin	154 (3.5)	<10 ^1^ (NA)
Urinary	286 (6.6)	24 (7.4)
Undefined	218 (5.0)	40 (12.3)

^1^ Due to Danish legislation of personal data protection, categories with less than 10 cases cannot be specified.

**Table 4 cancers-14-00284-t004:** Time-dependent hazard ratios of new primary cancer following the diagnosis of posterior uveal melanoma.

Years after Diagnosis of PUM	HR_t_ ^1^	95% CI ^2^	*p*-Value
0–1	1.10	(0.74; 1.63)	0.649
2–5	1.49	(1.23; 1.80)	<0.001
6–10	1.26	(0.99; 1.61)	0.058
11–15	1.49	(1.12; 1.99)	0.006
>15	1.27	(0.98; 1.65)	0.071

The model was adjusted for gender, age at diagnosis/index date and calendar year of diagnosis. ^1^ Time-dependent hazard ratio, ^2^ 95% confidence intervals.

## Data Availability

Collection and linkage of data was approved by the Danish Data Protection Agency (protocol no: 2016-41-4897) and the Danish Health Authority (protocol no: 3-3013-727). The study was based on data from the national administrative databases curated by Statistics Denmark. Access to the data is available after application to Statistics Denmark.

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
