# Peer review of "Risk of New Primary Cancer in Patients with Posterior Uveal Melanoma: A National Cohort Study"

_cancers, 2022, doi:10.3390/cancers14020284_

Round 1

Reviewer 1 Report

The authors investigated the risk of new primary cancer after PUM occurence on a cohort of 2,179 patients affected by PUM and 22,717 cancer-free controls. They found an increament of the incidence of newly-detected primary cancer, while no differences in the specific cancer types between controls and PUM patients were found.

The study is interesting and well written. The methods and the results also well presented.

I have some concerns:

  1. The authors shoudl expand the description of the conventional features of PUM. Genetic, clinical and radiologic features should be better mentioned. 
  2. The histopathology of PUM is not mentioned into the text. Use 10.3390/app10228081 and discuss it.

Author Response

Dear reviewer 1.

Thank you for taking the time to review our manuscript and for the relevant suggestions.

1) We have complied to your suggestion and expanded the basic description of uveal melanoma, please se the introduction section l.39-52

2) Discussion of the histopathology and the above-mentioned reference has been included in the introduction section, p.1 l. 39-41

Reviewer 2 Report

In the manuscript "Risk of New Primary Cancer in Patients with Posterior Uveal Melanoma. A National Cohort Study" by Mette Bagger et al. the authors conduct an innovative study based on the Danish population, comprising data collected over 5 decades, to determine whether patients diagnosed with posterior uveal melanoma are at an increased risk to develop a new different tumour type. The study is well conducted and the authors found that an association between posterior uveal melanoma and an increased risk of developing other cancers, irrespective of the social and economic status. These findings are extremely interesting and relevant for clinical practice, since they warrant an even more detailed follow-up of the patients diagnosed with uveal melanoma. In addition, the paper is extremely well written and organized. However, the paper could benefit from improvement in some details, which are the following:

  1. The authors put an emphasis on tumour predisposition related with BAP-1 germline mutations, but never mention other germline mutations that are associated with uveal melanoma development, such as, PALB2, MLH1 and MBD4. These mutation should be mentioned as well.
  2. In page 3, line 124, where it is written “diagnosed from 1,868 through to 2016”, it should be written “diagnosed from 1968 until 2016”.

Therefore, if considered relevant for publication in Cancers, the authors will have to address the minor aspects highlighted above before the final format of publication is achieved.

Author Response

Dear reviewer 2

Thank you for taking the time to review our manuscript and for the relevant suggestions.

1) We have mentioned the relevant mutations in the introduction and discussion section and provided a relevant reference.

p.2 l.60-62: "An association between hereditary predisposition to uveal melanoma and germline pathogenic variants in other known tumor genes such as MBD4, PALB2, SMARCE1 and MLH1 has also been suggested[10]"

p.9 l.250-252: "Whole exome sequencing of 27 patients with familial UM identified other potential pathogenic variants in PALB2 (1 case), SMARCE1 (1 case) and MLH1 (1case).[10] However, the role of these established cancer genes in uveal melanoma patients needs to be further investigated."

2) The typographical error has been corrected, thank you.

Reviewer 3 Report

However, the following are minor concerns:

  • There was a report that uveal melanoma cell line growth was regulated in-vitro by estrogen a female sex hormone. The fact that incidence in females was less compared to the incidence in males underlines this point. Hence, new primary cancers in females compared to males can be given separately in the table.
  • There is a typographical error (1868) under the results section in line no. 124.

Author Response

Dear reviewer 3

Thank you for taking the time to review our manuscript and for the relevant suggestions.

1) This is an interesting suggestion, however unfortunately we do not have the possibility of separating the incidence of new primary cancers on gender due to GDPR legislation, as number of cases will be less than five in several of the categories. Assessment of gender differences was beyond the scope of this study. However the the cox regression model is adjusted for potential gender related differences as gender is incorporated as covariate in the analysis.

2) The typographical error has been corrected, thank you